# Siderophore cheating and cheating resistance shape competition for iron in soil and freshwater *Pseudomonas* communities

Elena Butaitė[1], Michael Baumgartner[1], Stefan Wyder[1] & Rolf Kümmerli [1]

All social organisms experience dilemmas between cooperators performing group-beneficial actions and cheats selfishly exploiting these actions. Although bacteria have become model organisms to study social dilemmas in laboratory systems, we know little about their relevance in natural communities. Here, we show that social interactions mediated by a single shareable compound necessary for growth (the iron-scavenging pyoverdine) have important consequences for competitive dynamics in soil and pond communities of *Pseudomonas* bacteria. We find that pyoverdine non- and low-producers co-occur in many natural communities. While non-producers have genes coding for multiple pyoverdine receptors and are able to exploit compatible heterologous pyoverdines from other community members, producers differ in the pyoverdine types they secrete, offering protection against exploitation from non-producers with incompatible receptors. Our findings indicate that there is both selection for cheating and cheating resistance, which could drive antagonistic co-evolution and diversification in natural bacterial communities.

[1] Department of Plant and Microbial Biology, University of Zurich, Winterthurerstrasse 190, 8057 Zurich, Switzerland. Correspondence and requests for materials should be addressed to E.B. (email: elena.butaite@gmail.com) or to R.K. (email: rolf.kuemmerli@uzh.ch)

While microbes have become model organisms to study the evolution of cooperation in laboratory settings, we still know little about the role of microbial cooperative interactions in complex natural communities. Perhaps the most common form of microbial cooperation is the secretion of so-called public goods—compounds that are costly to produce but generate benefits for other cells in the vicinity of the producer[1]. Such public goods include matrix components to build up biofilms, enzymes to digest food, biosurfactants for cooperative swarming and iron-scavenging siderophores[1]. Many laboratory studies focused on the problem of cheating, a scenario where mutants that no longer contribute to public goods undermine cooperation by capitalising on the public goods secreted by others[2–6]. This body of work has become a paradigm for the public-goods dilemma, showing how a trait that is beneficial for the group can be selected against by the spread of selfish individuals[7]. While highly influential as a general proof of social evolution theory, a key open question is whether cheating and the public-goods dilemma also occur in natural microbial communities[8–10].

Here, we tackle this question by examining the potential for public-goods cooperation and cheating among pseudomonads from natural communities both at the genetic and behavioural level. *Pseudomonas* is a diverse genus of γ-proteobacteria, occupying a wide range of habitats (e.g. soil, aquatic ecosystems and animal hosts)[11]. Albeit diverse, many fluorescent pseudomonads share an important trait: they can produce and secrete pyoverdine, a siderophore that scavenges insoluble or host-bound iron from the environment[11]. Laboratory experiments have shown that pyoverdine is a public good that can be shared among cells, and be exploited by cheating mutants[2, 12]. Pyoverdine is a secondary metabolite produced via non-ribosomal peptide synthesis. The molecule consists of a conserved chromophore (making this molecule naturally fluorescent), an acyl side chain linked to the chromophore, and a variable peptide chain (6–12 amino acids)[13, 14]. The different *Pseudomonas* strains often produce slightly different pyoverdines, varying in the length and the composition of the peptide chain[15]. Moreover, while pyoverdine and its cognate receptor are typically specific, strains can also have less specific and/or several different receptors allowing the uptake of heterologous pyoverdines[16–19]. This pyoverdine-receptor diversity could facilitate different types of social interactions among co-occurring species. For instance, pyoverdine-producing strains could exploit each other's pyoverdines[19, 20]. Alternatively, pyoverdine non-producers could gain a foothold by exploiting foreign pyoverdines (i.e. could act as cheats)[21, 22]. Moreover, some strains might produce exclusive pyoverdine types, which remain inaccessible for competing non-isogenic strains because they lack a matching receptor—a scenario that could confer resistance to cheating[17, 23].

Although it has long been conjectured that the above-mentioned interactions could be important drivers of ecological and evolutionary dynamics in microbial communities[17, 22, 24–27], there is a lack of studies that have systematically examined siderophore-mediated social interactions, and the resulting fitness consequences among natural isolates in replicated communities. To date, the most comprehensive evolutionary study on siderophore-mediated interactions used a combination of whole-genome sequencing and phenotype screening to show that many marine *Vibrio* strains have lost the siderophore-synthesis cluster, but kept the receptor for uptake[26]. While this genomic pattern is compatible with the idea of siderophore non-producers being cheats, a direct demonstration of cheating during one-to-one strain competition is missing. Here, we build on the work by Cordero et al.[26] and demonstrate that: (a) pyoverdine non-producers co-occur with producers in soil and freshwater communities of *Pseudomonas*; (b) non-producers can exploit siderophores of certain community

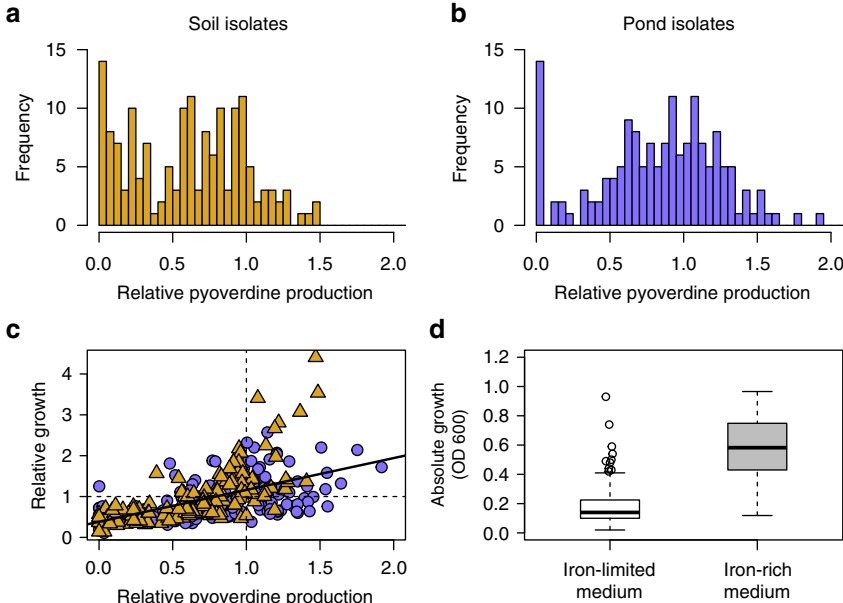

**Fig. 1** Pyoverdine production and growth of natural *Pseudomonas* isolates. The natural soil (**a**, $n = 158$) and pond (**b**, $n = 157$) isolates varied greatly in their levels of pyoverdine production. Pyoverdine production was measured in iron-limited casamino acids (CAA) medium, and was scaled relative to the production levels of characterised laboratory reference strains (Supplementary Table 1). **c** There was a significant positive correlation (*solid line*) between growth of the isolates in iron-limited media and their pyoverdine production level, suggesting that pyoverdine is important for growth. Values are given as means across three replicates for soil (*yellow triangles*) and pond (*purple circles*) isolates. *Dotted lines* denote growth and pyoverdine production of reference strains. **d** While the growth of natural isolates was strongly limited in media with low iron availability, all strains grew well when iron (40 μM FeCl$_3$) was added. This demonstrates that all natural isolates could use CAA as a nutrient source. *Box plots* show the median, first and third quartile, and the 95% confidence interval

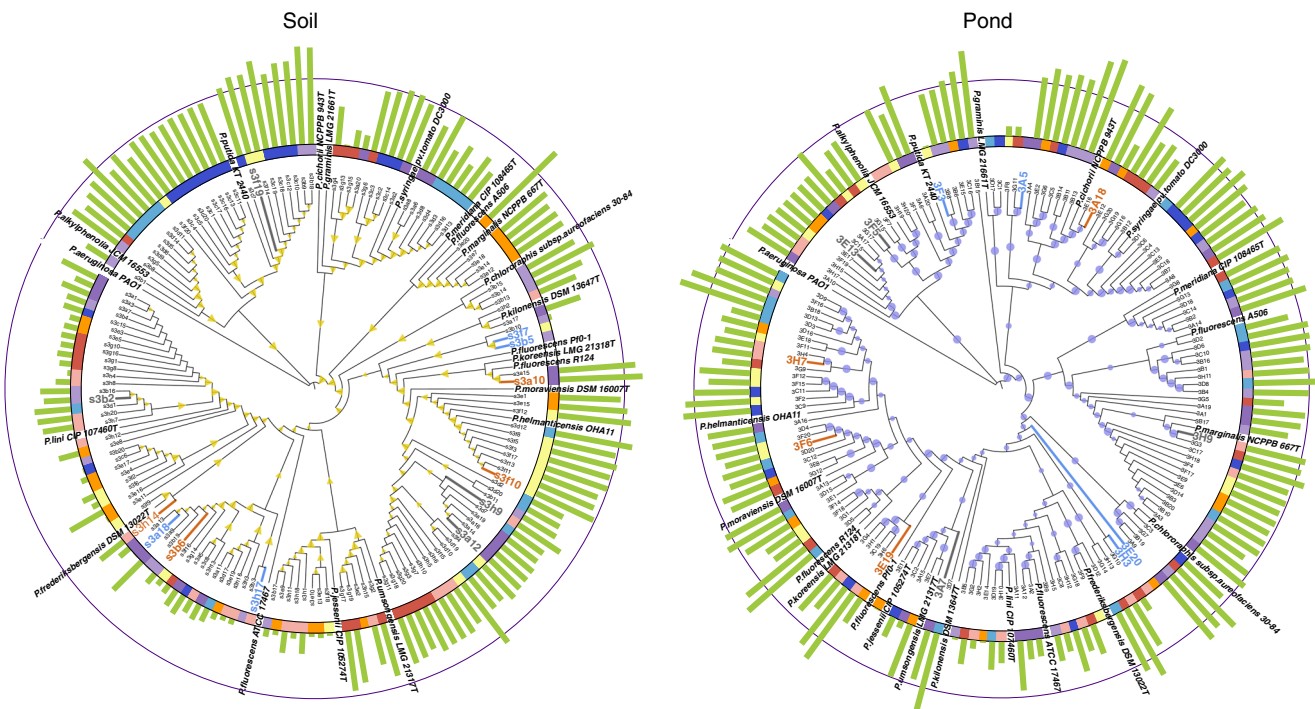

**Fig. 2** Maximum-likelihood cladograms for soil and pond isolates based on partial *rpoD* sequences. For both habitats, *P. aeruginosa* PAO1 was used as the outgroup, and published *rpoD* sequences of 20 members of the *P. fluorescens* lineage (*strain names in bold*) were integrated into the cladograms to demonstrate phylogenetic affiliation and diversity of our natural isolates. *Yellow triangles* and *purple circles* indicate bootstrap values (50–100%) for branches in the soil ($n = 148$) and the pond cladogram ($n = 149$), respectively. *Green bars* depict pyoverdine production levels of isolates relative to the laboratory reference strains (*grey ring*). *Colour strips* around cladograms represent the different communities from which isolates originated. IDs for strains used for pyoverdine cross-feeding and competition assays are *enlarged*, and the different colours depict pyoverdine non-producers (*light blue font*), producers with a growth-stimulatory pyoverdine (*orange font*) and producers with a non-growth-stimulatory pyoverdine (*grey font*)

members but not of others; (c) exploitation can lead to cheating, where non-producers gain a relative fitness advantage over producers in direct competition; (d) certain pyoverdines inhibit rather than promote the growth of non-producers; and (e) the patterns of cheating and growth inhibition can be explained by receptor compatibility and pyoverdine differences between strains at the molecular level. Taken together, our findings suggest that selection for cheating and resistance to cheating could spur antagonistic co-evolution and strain diversification in natural bacterial communities.

## Results

**Natural strains vary in their pyoverdine production levels**. We isolated 320 putative *Pseudomonas* strains from a total of eight soil and eight pond communities. For all isolates, we sequenced the *rpoD* gene (a commonly used phylogenetic marker for this genus[28]) to confirm that 315 isolates are pseudomonads. To investigate whether pyoverdine non-producers co-exist with producers in the same community, we measured the pyoverdine production profile of all 315 isolates in iron-limited casamino acids (CAA) medium. This assay revealed that strains producing no or residual amounts of pyoverdine (i.e. producing less than 5% compared to the laboratory reference strains listed in Supplementary Table 1) occurred in 14 of the 16 communities, with an overall abundance of 8.9% in both pond and soil communities. We further found that pyoverdine production is a continuous trait, with the production level ranging from zero to very high amounts (Figs. 1a, b). Variation in pyoverdine production levels was high in all communities (mean coefficient of variation, CV ± SE for soil communities: 61.8 ± 7.3%; for pond communities: 49.6 ± 3.4%; Supplementary

Fig. 1a), indicating that non-, low- and high-producer strains typically co-exist.

**Pyoverdine is important for growth under iron limitation**. Compatible with the view that pyoverdine is important for iron acquisition in these pseudomonads, we observed a significant positive correlation between the isolates' pyoverdine production levels and their growth in CAA medium, where iron was either bound to human apo-transferrin (Fig. 1c; linear mixed model (LMM): $t_{298} = 12.0$, $p < 0.001$) or the synthetic chelator 2,2′-dipyridyl (Supplementary Fig. 2a; LMM: $t_{298} = 19.67$, $p < 0.001$). To confirm that it is indeed the amount of pyoverdine that determines growth in iron-limited CAA and not the inability of certain strains to consume the provided nutrients, we further cultured the isolates in CAA supplemented with 40 μM $FeCl_3$. This experiment revealed substantial growth for all strains, showing that all environmental isolates can use CAA as a nutrient source (Fig. 1d, paired *t*-test comparing growth of each strain in iron-limited vs. iron-rich CAA: $t_{314} = -39.40$, $p < 0.001$).

**Linking pyoverdine production profiles to phylogeny**. To explore the relationship between phylogeny and pyoverdine production, we constructed maximum-likelihood phylogenetic trees based on partial *rpoD* gene sequences. We constructed separate trees for soil and pond habitats, and mapped pyoverdine production profiles of strains onto the trees (Fig. 2). The absolute phylogenetic diversity was significantly higher for pond than for soil communities (median ± (1st quartile | 3rd quartile) of Faith's phylogenetic diversity for soil communities: 1.43 ± (0.97 | 1.63); for pond communities: 2.49 ± (2.22 | 2.63); Mann–Whitney *U*-test: $W = 59$, $p = 0.003$). Conversely, the CV of phylogenetic

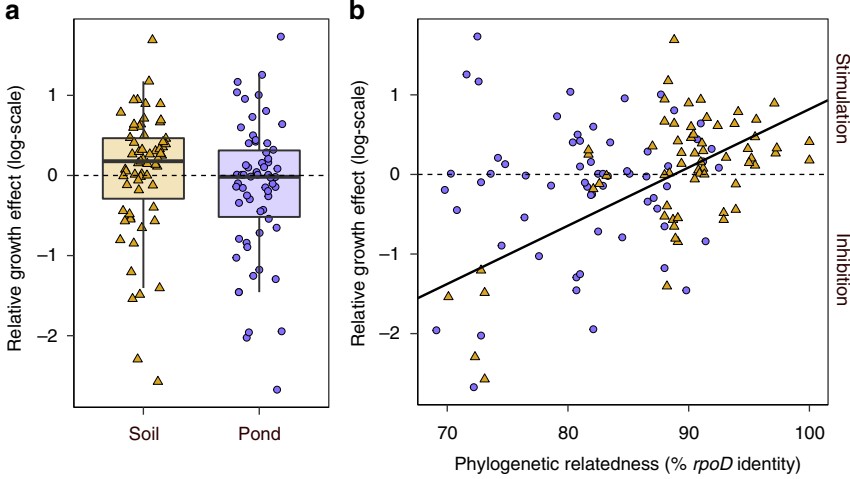

**Fig. 3** Foreign supernatants containing pyoverdine can have stimulatory, neutral or inhibitory effects on the growth of receiver strains. Supernatants containing pyoverdine from donor strains were fed to receiver strains from the same community, which produced lower amounts of pyoverdine than the donors. **a** The growth effect of foreign supernatants on receivers varied on a continuum from high inhibition to high stimulation. The average growth effect of donor supernatants on receivers did not significantly differ between soil (*yellow triangles*, n = 63 strain combinations) and pond (*purple circles*, n = 62) isolates. *Box plots* depict median, first and third quartile, and the 95% confidence interval. **b** There was a significant positive correlation (*solid black line*, linear mixed model: $t_{38} = 5.32$, $p < 0.001$) between the *rpoD* identity (i.e. relatedness) of receiver–donor pairs and the supernatant growth effect for soil, but not for pond isolates. The *dashed line* depicts neutral effects, whereas values greater or smaller than zero indicate cases of stimulation or inhibition, respectively, relative to conditions where receivers grew without donor supernatant. Data were ln-transformed prior to analysis

distance within community (a relative measure of diversity) did not differ between soil and pond, and was high for both habitats (mean CV ± SE for soil: 67.9 ± 6.6%; for pond: 58.9 ± 2.6%; *t*-test: $t_{14} = 1.26$, $p = 0.227$; Supplementary Fig. 1b). These analyses show that pseudomonads generally live in diverse, multi-strain communities.

We then examined whether there is a phylogenetic signal for pyoverdine production (i.e. whether closely related strains show similar production profiles). For this analysis, we calculated Blomberg's $K$ for each community, whereby, $K = 1$ indicates a phylogenetic signal as expected under the Brownian motion model of character evolution, and $K$-values close to zero stand for weak phylogenetic signals[29]. While we found a moderate phylogenetic signal for pond communities (mean ± SE across communities: $K = 0.254 ± 0.072$), there was no phylogenetic signal for pyoverdine production in soil communities ($K = 0.009 ± 0.005$; *t*-test for difference between habitats: $t_{14} = 3.37$, $p = 0.005$). These analyses highlight that pyoverdine production levels are not phylogenetically fixed, but highly vary even among closely related strains.

**Supernatants with pyoverdine affect growth of receivers.** We carried out a supernatant assay to test whether pyoverdine secreted by producers stimulates or inhibits the growth of community members producing no or little pyoverdine. From each community, we chose three strains producing no or low amounts of pyoverdine (henceforth called receivers) and fed them with the supernatant of three random strains (henceforth called donors), which produced higher amounts of pyoverdine than the receivers. Thus, each receiver was fed with pyoverdine-containing supernatants from three donors. This supernatant assay revealed that donor supernatants could both stimulate and inhibit the growth of receivers, with the effects varying on a continuum from complete inhibition to strong stimulation (Fig. 3a). The average effect of supernatants on growth did not significantly differ between soil and pond isolates (linear mixed model, LMM: $t_{14} = 0.731$, $p = 0.477$). Interestingly, the growth effect correlated positively with the phylogenetic relatedness

(based on *rpoD* sequences) between receivers and donors in soil but not in pond communities (Fig. 3b; LMM for pond: $t_{37} = 0.69$, $p = 0.492$; for soil: $t_{38} = 5.32$, $p < 0.001$). This result indicates that receivers from soil communities tended to be stimulated by more closely related strains. In pond communities, on the other hand, relatedness between receiver and donor pairs was generally lower than in soil (the highest receiver–donor *rpoD* identity was 92.5%, compared to 100% among soil isolates), and receivers were often stimulated by more distantly related donors (Fig. 3b).

The supernatant assays further revealed a significant donor effect (ANOVA: $F_{47,76} = 3.90$, $p < 0.0001$) and a receiver effect approaching significance (ANOVA: $F_{43,80} = 1.49$, $p = 0.062$). This means that supernatants of donors generally had consistent (either stimulating, neutral or inhibiting) effects on multiple receivers, and that receivers were often similarly affected by foreign supernatant regardless of the identity of the donor.

**Pyoverdine is responsible for the observed growth effects.** We then examined whether the above-reported stimulatory and inhibitory effects are indeed triggered by pyoverdine. To test this, we first randomly picked eight non-producers (four isolates were complete non-producers, i.e. relative pyoverdine production was indistinguishable from background fluorescence; and four isolates produced residual amounts of pyoverdine, i.e. 1.4–3.1% of laboratory reference strains; Table 1) from different soil and pond communities. We then grew each non-producer under iron-limited conditions with or without a purified pyoverdine from a producer from the same community, which showed a stimulatory effect on the non-producer in the supernatant assay (Fig. 3). We found a perfect match between the two assays (Fig. 4): all the eight pyoverdines isolated from strains previously shown to be stimulatory significantly promoted the growth of the non-producers. This suggests that these eight non-producers possess receptors to exploit the supplemented heterologous pyoverdines.

In a next assay, we fed the same eight non-producers with a purified pyoverdine from producers, which showed a neutral (two cases) or inhibitory (six cases) effect in the supernatant assay.

**Table 1 Characterisation of the pyoverdine locus and prediction of the pyoverdine amino acid backbone for natural Pseudomonas isolates**

| Community ID | Strain ID | Strain classification | Pyoverdine production (%)[a] | Number of genes | | | | | | Predicted pyoverdine peptide sequence[b,c] | |
|---|---|---|---|---|---|---|---|---|---|---|---|
| | | | | Regulator | Pyoverdine synthesis | Transport periplasma | Maturation periplasma | Secretion | Entire cluster | | |
| | | | | pvdS | pvdL pvdI pvdJ pvdD pvdH pvdA | pvdE | pvdQ pvdO pvdN pvdM pvdP | opmQ pvdR pvdT | | | |
| soil a | s3a18 | Residual non-producer | 2.8 | 1 | 5 | 1 | 5 | 3 | 15 | Lys-Orn-Gly-Gly-Thr-Ser-Orn | ● |
| | s3a10 | Stimulating producer | 127 | 1 | 5 | 1 | 5 | 3 | 15 | Ser-Orn-Gly-Gly-Thr-Gly-X | ■ |
| | s3a12 | Non-stimulating producer | 64 | 1 | 6 | 1 | 5 | 3 | 16 | Lys-Asp-Gly-Thr-Gly-Orn | ◆ |
| soil b | s3b5 | Residual non-producer | 1.4 | 1 | 6 | 1 | 5 | 3 | 16 | Ser-Lys-X-Thr-Ser-Orn | □ |
| | s3b6 | Stimulating producer | 20 | 1 | 5 | 1 | 5 | 3 | 15 | Lys-Orn-Gly-Gly-Thr-Ser-Orn | ● |
| | s3b2 | Non-stimulating producer | 42 | 1 | 7 | 1 | 5 | 3 | 17 | Gly-Orn-Gly-Gly-Ser-Gly-Asp-Thr | |
| soil f | s3f7 | Residual non-producer | 2.0 | 1 | 6 | 1 | 5 | 3 | 16 | Ser-Lys-X-Thr-Ser-Orn | □ |
| | s3f10 | Stimulating producer | 97 | 1 | 5 | 1 | 5 | 3 | 15 | Ser-Orn-Gly-Gly-Thr-Gly-X | ■ |
| | s3f19 | Non-stimulating producer | 120 | 1 | 7 | 1 | 5 | 3 | 17 | Asp-Lys-Asp-Ser-Thr-Gly-Thr-Lys-X | |
| soil h | s3h17 | Complete non-producer | 0.02 | 1 | – | – | 1 | – | 2 | N/A | |
| | s3h14 | Stimulating producer | 46 | 1 | 5 | 1 | 5 | 3 | 15 | Gly-Lys-Thr-Ser-X-Orn | ◇ |
| | s3h9 | Non-stimulating producer | 77 | 1 | 6 | 1 | 5 | 3 | 16 | Lys-Asp-Gly-Thr-Gly-Orn | ◆ |
| pond A | 3A5 | Complete non-producer | 0.001 | 1 | 1 | 1 | 1 | – | 4 | N/A | |
| | 3A18 | Stimulating producer | 139 | 1 | 7 | 1 | 5 | 3 | 17 | Lys-Asp-Thr-Thr-Gly-Asp-Ser | |
| | 3A7 | Non-stimulating producer | 90 | 1 | 5 | 1 | 5 | 3 | 15 | Gly-Lys-Thr-Ser-X-Orn-Thr-Thr | ◇ |
| pond E | 3E20 | Complete non-producer | 0.01 | 1 | – | – | 1 | 3 | 5 | N/A | |
| | 3E19 | Stimulating producer | 51 | 1 | 6 | 1 | 5 | 3 | 16 | Lys-Orn-Gly-Gln-Gly-Ser-Orn | |
| | 3E13 | Non-stimulating producer | 61 | 1 | 7 | 1 | 5 | 3 | 17 | Asp-Thr-Gly-Asp-Gln-Gln-Gly | |
| pond F | 3F3 | Residual non-producer | 3.1 | 1 | 7 | 1 | 4 | 3 | 16 | Asp-Lys-Orn-Thr-Gly-Ser-Ser-Orn | |
| | 3F6 | Stimulating producer | 63 | 1 | 6 | 1 | 5 | 3 | 16 | Lys-Asp-Gly Thr-Gly-Orn | ◆ |
| | 3F5 | Non-stimulating producer | 48 | 1 | 7 | 1 | 5 | 3 | 17 | Gly-X-X-Asp-X-Orn | |
| pond H | 3H3 | Complete non-producer | 0.01 | 1 | 1 | 1 | 1 | 3 | 7 | N/A | |
| | 3H7 | Stimulating producer | 72 | 1 | 6 | 1 | 5 | 3 | 16 | Lys-Asp-Gly-Thr-Gly-Orn | ◆ |
| | 3H9 | Non-stimulating producer | 108 | 1 | 6 | 1 | 5 | 3 | 16 | Ser-Lys-Thr-Ser-X-Orn-Thr-Thr-X | |

[a]Pyoverdine production (in percentage) was measured relative to the reference strains (Supplementary Table 1)
[b]X = unknown amino acid (no significant hit with prediction software)
[c]Symbols depict pyoverdines with identical peptide backbone composition

In five cases results were consistent across the two assays: pyoverdines isolated from strains previously shown to be inhibitory significantly compromised the growth of the non-producers. Although the match was not perfect in the remaining three cases (Fig. 4, community a and b: neutral effect in the supernatant assay vs. inhibition in the pyoverdine cross-feeding assay; community A: inhibitory effect in the supernatant assay vs. neutral effect in the pyoverdine cross-feeding assay), pyoverdine never had a stimulatory effect. These findings strongly suggest that these non-producers lack receptors for the uptake of this second batch of pyoverdines. In this scenario, growth suppression can arise because incompatible pyoverdines lock away iron in the media, thereby further reducing the availability of this essential element for non-producers.

**Cheating and cheating resistance in direct competition.** The above findings indicate that it is mainly pyoverdine that drives the interaction patterns between our natural isolates

under iron limitation. Consequently, we sought to understand how pyoverdine-mediated growth effects (ranging from stimulation to inhibition) impact the competitive abilities of strains. Accordingly, we carried out 16 direct pairwise competition assays where we mixed the eight non-producers with either their stimulating or non-stimulating (neutral or inhibitory) pyoverdine producers. To be able to distinguish the two competing strains, we integrated a constitutively expressed, fitness-neutral, mCherry marker into the chromosome of the non-producers (Supplementary Fig. 3).

When grown as monocultures, the non-producers grew significantly worse than the producers (Fig. 5a, paired $t$-test for non-producers vs. stimulating producers: $t_7 = -3.87$, $p = 0.006$; non-producers vs. inhibiting producers: $t_7 = -9.64$, $p < 0.001$). These growth patterns indicate that it is a handicap to be a pyoverdine non-producer in iron-limited medium. In contrast, fitness patterns reversed in direct competition for four strain pairs, where the non-producers could significantly outcompete their stimulating producers (Fig. 5b). This finding strongly

suggests that non-producers can act as cheats by successfully exploiting the pyoverdine secreted by producers, thereby gaining a relative fitness advantage. However, our competition assays also revealed that the ability to use a heterologous pyoverdine is not necessarily enough to gain a relative fitness advantage, as evidenced by the four cases where the non-producers lost in competition against producers secreting a compatible pyoverdine (Fig. 5b). Finally, we found that the non-producers performed worse and were strongly outcompeted in co-cultures with producers secreting an incompatible pyoverdine (Fig. 5b).

**The genetic basis of pyoverdine-mediated social interactions.** The results from the pyoverdine cross-feeding and competition assays suggest that: (a) the growth-stimulating and non-stimulating pseudomonads from the same community produce different pyoverdines; (b) the non-producers have receptors for heterologous pyoverdine uptake; and (c) the pyoverdine receptors of the non-producers are more similar to the receptors of the stimulating than the non-stimulating producers. To test these hypotheses, we sequenced the whole genome of the 24 strains used in the pyoverdine cross-feeding and competition assays (Figs. 4 and 5).

We first compared the organisation of the pyoverdine locus of each strain to that of previously characterised pseudomonads[14, 17, 30]. We found that each producer has a single complete pyoverdine locus, consisting of genes encoding the iron-starvation sigma factor *pvdS*, the pyoverdine-synthesis machinery (i.e. the non-ribosomal peptide synthesis assembly line), the export elements required for secretion, and a receptor (i.e. a *fpvA* homologue) for uptake (Table 1). In contrast, the four complete non-producers have a highly truncated pyoverdine cluster, where large genomic regions coding for the synthesis machinery are missing. In contrast, *pvdS* and the receptor gene are still present in these strains (Table 1). The four strains producing residual amounts of pyoverdine, meanwhile, all have a complete pyoverdine cluster (Table 1). The reason why the latter strains are unable to produce wild type amounts of pyoverdine must therefore reside in alterations of regulatory elements, as was found to be the case for evolved *P. aeruginosa* pyoverdine non-producers[31, 32]. Because the exact pyoverdine regulon is unknown for our natural isolates, it was impossible to directly test this hypothesis. In summary, our assembly analysis revealed that there are two types of non-producers: structural non-producers with a truncated pyoverdine locus and silent non-producers with a complete, yet largely inactive locus.

To investigate hypothesis (a), we used the non-ribosomal peptide synthesis assembly line identified in the producers to predict the peptide sequence of the pyoverdine backbone. These analyses indicate that the stimulating and the non-stimulating producers, from the same communities, produce structurally different pyoverdines (Table 1), which means that each non-producer was indeed confronted with two different types of pyoverdines in our assays. When focusing on pyoverdine uptake, we found that all the non-producers have multiple homologues of the *fpvA* receptor in their genome (Table 2). This finding supports hypothesis (b), as it shows that the non-producers (but also the producers) seem to be equipped for taking up heterologous pyoverdines. Given the many *fpvA* homologues, the direct testing of hypothesis (c) becomes difficult, as we do not know which homologue is actually used to take up the specific pyoverdine provided in our assays. Nonetheless, we performed two comparisons that serve as proxies for testing hypothesis (c). In the first of these, we compared the sequence similarity of FpvA encoded in the producer's pyoverdine locus to any FpvA homologues in the non-producer. In the second

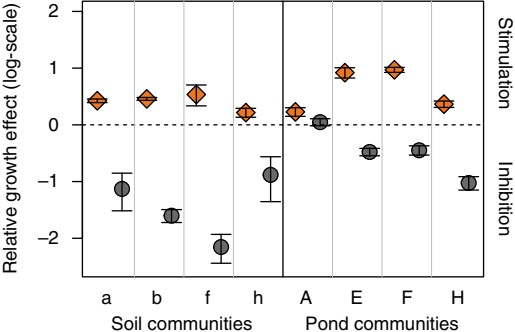

**Fig. 4** Heterologous pyoverdines promote or inhibit growth of receiver strains. Eight pyoverdine non-producers, each originating from a different community, were supplemented with purified pyoverdine from either (i) a strain, which showed a stimulatory effect (*orange diamonds*) in the supernatant assay, or (ii) a strain, which showed neutral (two cases) or inhibitory (six cases) effect (*grey circles*) in the supernatant assay. The *dashed line* depicts neutral effects, whereas values greater or smaller than zero indicate cases of growth stimulation or inhibition, respectively, relative to conditions where receivers grew without heterologous pyoverdine. The data show a good match between the supernatant and the pyoverdine cross-feeding assays, suggesting that it is indeed heterologous pyoverdine that triggers the patterns of growth stimulation and inhibition reported in Fig. 3. Data points show means ± 95% confidence intervals across five replicates. Data were ln-transformed prior to analysis

comparison, we analysed the similarity of the FpvA sequence encoded in the pyoverdine locus (in residual non-producers) or close to its remains (in complete non-producers) to any of the FpvA homologues in the producers. Both comparisons were in support of hypothesis (c): receptor similarities were higher between the non-producers and their corresponding stimulating producers than between the non-producers and the non-stimulating producers (comparison 1: respective FpvA similarities (mean ± SE) were $0.58 \pm 0.05$ vs. $0.40 \pm 0.01$, paired *t*-test: $t_7 = 3.61$, $p = 0.009$; comparison 2: respective FpvA similarities were $0.79 \pm 0.05$ vs. $0.48 \pm 0.11$, paired *t*-test: $t_7 = 3.15$, $p = 0.016$; Table 2).

## Discussion

Cheating is characterised by individuals exploiting the benefits of cooperative acts performed by others[33]. This phenomenon has been extensively studied in microbial laboratory systems in the context of fruiting body[34, 35] and biofilm[6, 36] formation, group defence strategies[37, 38], swarming motility[5, 39], enzyme[3, 4], toxin[40] and siderophore[2, 12, 41–43] production. Although social interactions and cheating seem to cover all aspects of microbial life, their role in natural microbial communities remains largely unclear (apart from fruiting body formation)[44–47]. Our study tackled this gap in knowledge and shows that social interactions mediated by shareable iron-scavenging pyoverdines can have important consequences for strain-to-strain interactions in phylogenetically diverse natural soil and pond *Pseudomonas* communities. In particular, we found that: (a) strains that produce no or low amounts of pyoverdine commonly occur in natural communities, although iron is a key growth-limiting factor for pseudomonads in natural habitats[11]; (b) there are two types of non-producers, which both likely evolved from ancestral producers: structural non-producers with a truncated pyoverdine locus and silent non-producers with a complete, yet largely inactive locus; (c) the non-producers possess multiple pyoverdine receptors, and they are able to capitalise on

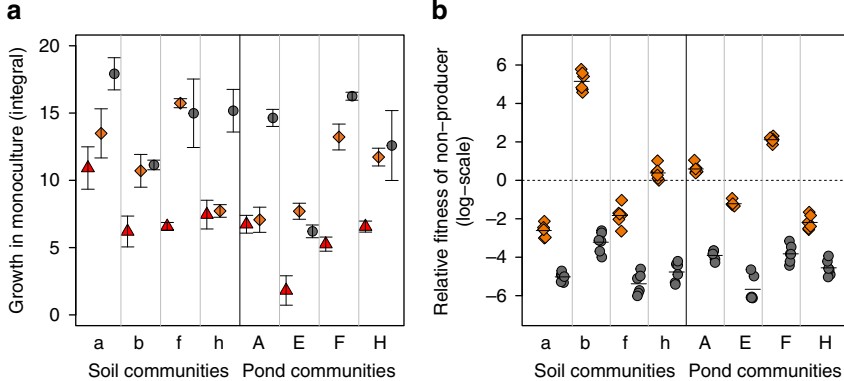

**Fig. 5** Direct competition between pyoverdine producers and non-producers reveal both cheating behaviour and resistance to cheating. **a** When grown as monocultures, the eight pyoverdine non-producers grew significantly worse than the pyoverdine producers, showing that the inability to produce pyoverdine represents a handicap in iron-limited CAA medium. Growth of non-producers (*red triangles*) tagged with a neutral constitutive mCherry marker was compared to growth of the strains producing a pyoverdine type that can (*orange diamonds*) or cannot (*grey circles*) be used by the respective non-producers for iron uptake. Data points show means ± 95% confidence intervals across three replicates. **b** Direct competition between non-producers and producers revealed three different scenarios. When competing against producers that secrete a compatible pyoverdine (*orange diamonds*), non-producers could win the competition in four cases. In the other four cases, non-producers could not outcompete producers despite being able to use the producer's pyoverdine. Finally, when competing against producers that secrete an incompatible pyoverdine (*grey circles*), non-producers were always strongly outcompeted. The *dashed line* depicts fitness parity, whereas values greater or smaller than zero indicate cases where non-producers or producers won the competition, respectively. While symbols show individual data points, bars depict means across six replicates. Data were ln-transformed prior to analysis

pyoverdine produced by other community members; (d) certain non-producers can outcompete producers from the same community in direct competition through pyoverdine exploitation; and (e) some producers secrete incompatible pyoverdines that repress rather than promote the growth of non-producing community members. These results highlight that siderophores play an important, multi-faceted role in shaping social interactions between co-occurring bacterial strains, and are likely to drive diversification and competitive dynamics in natural bacterial communities.

Our study revealed that pyoverdine-mediated social interactions among natural pseudomonads are complex, and not only involve cheating, as observed for laboratory strains[2, 12, 41], but also include pyoverdine-mediated growth inhibition, and the use of heterologous pyoverdine that results in an absolute but not a relative fitness benefit for non-producers (Figs. 3–5). One reason for this increased complexity is that social interactions in our natural communities occur among phylogenetically diverse strains (Fig. 2), which differ in the pyoverdines they produce and the number and types of receptors they possess (Tables 1–2). For instance, our observation that certain non-producers can exploit and outcompete producers suggests that these non-producers possess a high-affinity receptor for specific heterologous pyoverdines. Conversely, our observation that certain non-producers are inhibited by other heterologous pyoverdines suggests that these non-producers possess incompatible receptors, which excludes them from social interactions, and thus allows producers to privatise pyoverdine and iron uptake[48]. Finally, our finding that some non-producers can exploit heterologous pyoverdines, but do not outcompete the producers, could indicate that these non-producers have receptors with relatively low affinity for these particular pyoverdines, or that non-producers were kept in check by producers through other mechanisms, such as toxin-mediated interference competition[49].

Another source of complexity is our observation that pyoverdine production is a continuous and not a binary trait, as it is typically the case in laboratory experiments where knockout mutants are used as non-producers. Genetic work on laboratory *P. aeruginosa* strains revealed that point mutations in the regulatory gene *pvdS* can lead to continuous variation in PvdS

activity and levels of pyoverdine production[31, 50]. The consequence of this is that not only non-producers can exploit producers, but potentially any strain that produces a lower amount of pyoverdine than its competitor could act as a cheat[51–53]. Moreover, the continuous nature of the pyoverdine trait could favour facultative cheating[54], where strains invest to some extent in their own pyoverdine production when growing alone, but switch to the exploitation of heterologous pyoverdines when other strains are nearby. Our genomic analysis supports the idea of strains exhibiting flexible facultative strategies, as most sequenced isolates possess multiple different pyoverdine-receptor homologues (median 4, range 1–19; Table 2). Moreover, strains like those four that had a complete pyoverdine locus, yet only produced residual amounts of pyoverdine (Table 1) could be candidates pursuing facultative strategies: sustain themselves with the little amount of pyoverdine they make when growing alone, but switch to exploitation in co-culture with other producers.

We might now ask what the consequences of the reported social interactions for the long-term evolutionary dynamics in natural communities might be. Previous work proposed that one way to escape cheating is to mutate pyoverdine and receptor types[17, 25, 27]. This evolutionary response of producers could in turn impose selection on non-producers to mutate their receptors accordingly, to acquire new compatible receptors through horizontal gene transfer or to evolve broad-range receptors. These evolutionary adaptations and counter-adaptations could infinitely continue and lead to antagonistic co-evolution generating ever new variants of pyoverdines and receptors[17, 31]. Several of our findings are in line with the scenario of antagonistic co-evolution: we observed a high pyoverdine diversity among producers (i.e. the 16 producers analysed in Table 1 produced 11 different pyoverdine types). Furthermore, we found that many strains possess multiple pyoverdine receptors (Table 2), and non-producers could only use the pyoverdine of certain producers but not of others (Fig. 4). Finally, some producers were consistently resistant against exploitation by multiple non-producers, whereas other producers were particularly vulnerable to exploitation. All these findings together indicate that some strains might be ahead of the evolutionary race, by

**Table 2 Numbers and similarities of the FpvA pyoverdine receptors between producers and non-producers**

| Community ID | Strain ID | Strain classification | Number of fpvA homologues | Highest similarity of the FpvA in the producer's pyoverdine locus to the FpvA homologues in the non-producer | | Highest similarity of the FpvA in the non-producer's pyoverdine locus to the FpvA homologues in the producer |
| --- | --- | --- | --- | --- | --- | --- |
| | | | | Stimulator | Non-stimulator | Non-producer |
| soil a | s3a18 | Residual non-producer | 2 | 0.76 | 0.36 | 1.00 |
| | s3a10 | Stimulating producer | 1 | 1.00 | – | 0.84 |
| | s3a12 | Non-stimulating producer | 2 | – | 1.00 | 0.83 |
| soil b | s3b5 | Residual non-producer | 4 | 0.44 | 0.38 | 1.00 |
| | s3b6 | Stimulating producer | 2 | 1.00 | – | 0.92 |
| | s3b2 | Non-stimulating producer | 2 | – | 1.00 | 0.88 |
| soil f | s3f7 | Residual non-producer | 4 | 0.46 | 0.41 | 1.00 |
| | s3f10 | Stimulating producer | 5 | 1.00 | – | 0.89 |
| | s3f19 | Non-stimulating producer | 11 | – | 1.00 | 0.23 |
| soil h | s3h17 | Complete non-producer | 3 | 0.74 | 0.36 | 1.00 |
| | s3h14 | Stimulating producer | 4 | 1.00 | – | 0.89 |
| | s3h9 | Non-stimulating producer | 2 | – | 1.00 | 0.83 |
| pond A | 3A5 | Complete non-producer | 5 | 0.50 | 0.42 | 1.00 |
| | 3A18 | Stimulating producer | 4 | 1.00 | – | 0.62 |
| | 3A7 | Non-stimulating producer | 4 | – | 1.00 | 0.24 |
| pond E | 3E20 | Complete non-producer | 7 | 0.45 | 0.37 | 1.00 |
| | 3E19 | Stimulating producer | 3 | 1.00 | – | 0.84 |
| | 3E13 | Non-stimulating producer | 19 | – | 1.00 | 0.24 |
| pond F | 3F3 | Residual non-producer | 10 | 0.59 | 0.41 | 1.00 |
| | 3F6 | Stimulating producer | 2 | 1.00 | – | 0.51 |
| | 3F5 | Non-stimulating producer | 8 | – | 1.00 | 0.37 |
| pond H | 3H3 | Complete non-producer | 9 | 0.69 | 0.47 | 1.00 |
| | 3H7 | Stimulating producer | 3 | 1.00 | – | 0.83 |
| | 3H9 | Non-stimulating producer | 6 | – | 1.00 | 0.25 |

either being particularly successful in heterologous pyoverdine exploitation or by being generally resistant to it.

For a complete understanding of the system, we would need to know how the pairwise interactions investigated in our study add up at the community level. While a definite answer is not yet possible, a recent laboratory study examined population dynamics in communities where non-producers simultaneously interacted with one producer secreting a compatible pyoverdine, and another producer secreting an incompatible pyoverdine[27]. This study revealed non-transitive competitive dynamics, where non-producers outcompeted producers with a compatible pyoverdine, but were themselves outcompeted by producers with an incompatible pyoverdine. Overall, strains chased each other in a competitive race with no overall winner, which resulted in the maintenance of biodiversity and stable community composition. Our data now reveal that cheating non-producers and cheating-resistant producers are indeed both present in our communities, which opens the possibility for these biodiversity-promoting mechanisms to operate in natural systems.

How does our work compare to the seminal study by Cordero et al.[26] who showed that siderophore non-producing and producing *Vibrio* strains co-exist with one another in a marine ecosystem? One important insight from our study is that social interactions between producers and non-producers are not restricted to marine communities, but also occur in soil and freshwater ecosystems, in a completely different taxon. This highlights that siderophore-mediated interactions between taxonomically diverse strains are likely a common feature of microbial communities. Another similarity between the two studies is that there is no strong phylogenetic signal for siderophore production (Fig. 2). This means that pyoverdine non- or low producers were not limited to a few specific taxonomic clades, but occurred across the entire phylogenetic tree. This indicates that, as for the *Vibrio* system, non-producers frequently arise de novo from within producer clades. However, the mechanism by which non-producers evolve differs between the *Vibrio* and our *Pseudomonas* system. Particularly, Cordero et al.[26] found discrete phenotypes: strains were either full producers (40%) or non-producers (60%), and these phenotypes correlated well with the presence or absence of the siderophore-synthesis clusters in the genome of these strains. Conversely, in our system non-producers are relatively rare (9%) and come in two different forms: structural non-producers with a truncated pyoverdine locus and silent non-producers with a complete, yet largely inactive locus, producing only residual amounts of pyoverdine. In addition, isolates showed continuous phenotypes, from pyoverdine non- to full production (Fig. 1). This demonstrates that most isolates have not lost their pyoverdine-synthesis cluster, and suggests that modifications in regulatory elements might rather be the key determinants of how much pyoverdine a strain is capable to produce.

Another key difference between the two study systems is that structural diversity exists for pyoverdine (Table 1), whereas the

*Vibrio* siderophores (aerobactin and vibrioferrin) come in a single molecular form[55]. Because pyoverdine diversity could select for receptor diversity[17, 18], successful cheating is then not so much about having a receptor per se, but rather about having a matching receptor (Table 2). We can think of two scenarios of how matching receptors can be acquired. First, if non-producers evolve de novo from producers, then they inherently possess the matching receptor of the producer they originated from. This route to exploitation might commonly apply in spatially structured habitats where de novo non-producers can rely on closely related producers staying in close vicinity. Indeed, our supernatant assays suggest that pyoverdine-mediated growth stimulation preferentially occurs among closely related strains in soil, a highly structured environment (Fig. 3b). Second, non-producers could acquire matching receptors through horizontal gene transfer. This route to exploitation might preferentially occur in habitats with low spatial structure, where strains readily mix and closely related producers are not necessarily nearby. This scenario indeed seems to apply to our pond communities, living in a fairly unstructured habitat, where pyoverdine-mediated growth stimulation primarily occurred among more distantly related strains (Fig. 3b).

In conclusion, our findings demonstrate that pyoverdine-mediated cheating and competition for iron are prevalent among natural *Pseudomonas* isolates and have important fitness consequences. Because iron scavenging via siderophores is ubiquitous among bacterial taxa in iron-limited habitats[11, 55], we propose that siderophore-mediated social interactions are important in many ecosystems, and are likely involved in shaping strain diversity and community dynamics. We further propose that other microbial social traits might play similar roles. For instance, many bacterial species secrete small signalling molecules (e.g. acyl homoserine lactones, AHLs) for communication and the coordination of group-level activities[56]. It has previously been shown that AHLs are exploitable public goods[3, 4], are structurally diverse and occur in many different species[57]. The complex social interactions uncovered here for siderophores might thus well apply to AHL-based communication systems. Taken together, our study highlights that not only abiotic but also social components need to be considered in order to fully understand microbial community assembly and functioning.

## Methods

**Sampling and isolation of pseudomonads**. We sampled 16 *Pseudomonas* communities from soil and pond habitats ($n = 8$ each) located on the campus of the University of Zurich Irchel (47.40° N, 8.54° E), Switzerland. We used the following sampling and isolation protocol, adapted from previous studies[58–60]. For soil sampling, we used a metal soil probe with a 7 mm diameter slot to sample the upper 10 cm of the soil. From the extracted soil cores, we discarded the upper and the lower 2 cm, and processed the middle part for strain isolation. For pond sampling, we collected water close to the shore from the upper layer by totally immersing 250 ml sterile glass bottles into the water. All samples were transported on ice and processed in the laboratory within 25 h. For the soil samples, approximately 1 g was suspended in 9 ml of 0.85% NaCl solution, and then vortexed vigorously for 2 min. We then plated 400 μl of $10^{-1}$, $10^{-2}$ dilutions (in 0.85% NaCl solution) of the bacteria suspensions on agar plates containing Gould's S1 medium supplemented with 100 μg/ml of the antifungal cycloheximide and 50 μM FeCl$_3$ (to also allow potential siderophore non-producers to grow). This medium is selective for fluorescent *Pseudomonas*[61]. The pond samples were filtered through 0.22 μm PES bottle-top vacuum filter (Millipore steritop-GP 250 ml). A stirring rod and 2.5 ml 0.85% NaCl solution were added to the filter, sealed at the bottom with parafilm, and the filter with suspension was stirred for 3 min. We plated 240 μl of 0, $10^{-1}$ dilutions of the bacteria suspensions on Gould's S1 medium. Plates were incubated for three days at room temperature in the dark. After incubation, we picked 20 random isolates per community and streaked them out on lysogeny broth (LB) agar plates to finally isolate a single purified colony (320 in total). These isolates were grown statically in 1 ml LB for 24–48 h in 24-well plates. Glycerol was added to a final concentration of 25% to prepare freezer stocks for storage at −80 °C. Each isolate obtained an identification code, consisting of

location ID ('s3': soil Irchel campus; '3': pond Irchel campus), followed by a letter (for soil communities: small letters 'a' to 'h'; for pond communities: capital letters 'A' to 'H') and a number (1–20).

**rpoD amplification and sequencing**. To verify that the isolates are indeed pseudomonads, we PCR amplified and sequenced a part of the housekeeping *rpoD* gene for 315 isolates (PCR or sequencing failed for 5 isolates, which were excluded from further experiments). *rpoD* is commonly used for phylogenetic affiliation of pseudomonads[28, 62]. PCR mixtures contained 2 μl of 10 μM solutions of each primer, PsEG30F and PsEG790R[28], 25 μl Quick-Load Taq 2X Master Mix (New England Biolabs), and 21 μl of sterile Milli-Q water. We added bacterial biomass either from single colonies on LB plates or directly from glycerol stocks (note that the latter method worked much better than the former one) to the PCR mixture distributed in 96-well PCR plates. Plates were sealed with an adhesive film. The following PCR conditions were used: denaturation at 94.5 °C for 10 min; 30 cycles of amplification (1 min denaturation at 94 °C, 1 min primer annealing at 55 °C, and 1 min primer extension at 68 °C); final elongation at 68 °C for 10 min (adapted from ref. [28]). The PCR products were purified and commercially sequenced using the PsEG30F and/or the PsEG790R primer.

**Taxonomic and phylogenetic affiliation**. We used NCBI BLASTN analysis for taxonomic assignment of *rpoD* sequences. To cover the broad species diversity of the isolates, we chose 20 characterised *Pseudomonas* all belonging to the *P. fluorescens* lineage as reference strains for phylogenetic trees (Supplementary Table 2): four main groups (*P. fluorescens*, *P. lutea*, *P. putida* and *P. syringae*), and seven subgroups of the *P. fluorescens* group (*P. chlororaphis*, *P. corrugata*, *P. fluorescens*, *P. gessardi*, *P. jessenii*, *P. koreensis* and *P. mandelii*)[63, 64]. *P. aeruginosa* PAO1 was used as an outgroup.

We retrieved partial or full sequences of all reference and outgroup strains from GenBank. We used MEGA 7 software (Supplementary Table 3) to manipulate *rpoD* sequences and construct phylogenetic trees. *rpoD* sequences of all strains were converted to amino acid sequences, and aligned with MUSCLE. We checked quality of alignments manually and removed the shortest sequences (to guarantee a sequence length ≥ 600 bp), leaving us with 297 *rpoD* sequences of natural isolates. Sequences in the alignment were trimmed at both ends to obtain maximum overlap and reconverted to nucleotide sequences, resulting in 609 bp sequences for all strains. We constructed maximum-likelihood (ML) trees, using General time reversible (GTR) + G + I model, which yielded the best fit to our data set. Bootstrapping was carried out with 100 replicates, keeping gaps. We displayed and manipulated ML trees using the iTOL web tool (Supplementary Table 3). For some analyses, we calculated the relatedness between strains by carrying out pairwise alignments of *rpoD* sequences using EMBOSS Water (Supplementary Table 3).

**Measurement of growth and pyoverdine production levels**. To evaluate whether natural isolates can produce pyoverdine, we grew all isolates under iron-limited conditions and assessed their pyoverdine production levels. We first grew isolates in 150 μl LB in 96-well plates overnight (16–18 h) under static conditions at room temperature. We then transferred 2 μl of overnight cultures to 200 μl iron-limited CAA medium (containing 5 g casamino acids, 1.18 g K$_2$HPO$_4$·3H$_2$O, 0.25 g MgSO$_4$·7H$_2$O per litre) supplemented with 25 mM HEPES buffer, 20 mM NaHCO$_3$ and 100 μg/ml human apo-transferrin (a strong natural iron chelator) in a 96-well plate. All chemicals were purchased from Sigma-Aldrich, Switzerland. Each plate contained isolates from one community in triplicates and eight reference strains from our strain collection known to produce pyoverdine (Supplementary Table 1). Following 18 h of incubation at room temperature in the dark, we measured growth (optical density OD at 600 nm) and pyoverdine production levels (relative fluorescence units (RFU) with excitation: 400 nm and emission: 460 nm) with an Infinite M200 Pro microplate reader (Tecan Group Ltd., Switzerland)[65]. We then calculated the relative growth and pyoverdine production for each isolate by dividing its OD$_{600}$ and RFU by the average respective OD$_{600}$ and RFU of the reference strains.

We carried out control experiments to verify whether it is indeed iron limitation that induces the observed growth and pyoverdine production patterns. First, we checked whether all isolates were able to use CAA as a nutrient source by growing strains in the same way as described above, just this time with 40 μM FeCl$_3$ supplemented to CAA (Fig. 1d). Second, to rule out specific effects in response to transferrin as iron chelator we repeated the growth experiments with a synthetic iron chelator, 2,2′-dipyridyl (400 μM, 24 h growth assay). These experiments yielded qualitative similar results for both iron chelators (Supplementary Fig. 2). Finally, we verified that end point OD$_{600}$ is a reliable measure of growth by correlating this measurement to growth parameters extracted from time-course data, and by relating end point OD$_{600}$ to actual cell counts obtained from flow cytometry (Supplementary Fig. 4). Data on growth trajectories were obtained by growing a subset of isolates ($n = 155$) in 200 μl CAA with 400 μM 2,2′-dipyridy in 96-well plates in a Tecan microplate reader at room temperature for 24 h. OD$_{600}$ was measured every 30 min prior a 15-second shaking event (3 mm orbital displacement). Some of the natural isolates exhibited non-standard growth trajectories, characterised by a long linear rather than an exponential growth phase. Because such patterns prevent the fitting of parametric growth models, we applied

non-parametric spline fits using the R 'grofit' package (Supplementary Table 3). From these spline fits, we extracted the integral (area under the curve) as the growth parameter of interest. The integral is a representative growth measure as it combines information on the lag, growth and stationary phase in one single estimate[66]. Flow cytometry was carried out on a subset of soil ($n = 24$) and pond ($n = 24$) isolates grown in CAA medium with 200 μM 2,2-dipyridyl in a 96-well plate for 24 h. $OD_{600}$ was measured in a Tecan microplate reader and samples were subsequently diluted 100× in 0.85% NaCl solution. Cells were fixed with glutaraldehyde (final concentration 2.5%, Sigma-Aldrich, Switzerland) and stained with Sybr Green I ($5 \times 10^{-5}$ dilution of commercial stock, Invitrogen, USA) for 10 min at room temperature in the dark. Samples were analysed using an InFlux V-GS cell sorter (Becton Dickinson Inc., USA). A blue laser (200 mW, 488 nm) was used for detection of side-scattered (SSC) light and Sybr Green I fluorescence (531 nm). Analysis of flow cytometry data was carried out with an in-house custom software (J. Villiger and J. Pernthaler, University of Zurich, unpublished) and bacterial cells were determined using SSC vs. green fluorescence. These control experiments revealed highly significant correlations between end point $OD_{600}$, growth integrals and cell counts demonstrating that using end point $OD_{600}$ is a reliable measure of growth for the natural isolates (Supplementary Fig. 4).

**Supernatant assay.** For each community, we harvested supernatants from three pyoverdine-producing isolates (donors) and fed them to three receiver strains. Receivers were always the three isolates with the lowest pyoverdine production levels in the community (mean ± SE relative pyoverdine production levels of receivers: 0.183 ± 0.038, $n = 44$). In all but one case (pond community G) all three receivers differed in their $rpoD$ sequence, and thus represent phylogenetically different strains. For donors, we picked three random isolates, which had higher pyoverdine production levels than the receivers (mean ± SE relative pyoverdine production levels of donors: 0.785 ± 0.056, $n = 48$).

To generate pyoverdine-containing supernatants, we grew isolates in 4 ml CAA medium with 200 μM 2,2′-dipyridyl in 14 ml polypropylene round-bottom tubes, shaken (160 rpm) at 28 °C. Supernatants were isolated in late exponential phase ($OD_{600} = 0.3$–0.5; measured with Tecan microplate reader), and centrifuged for 2 min at 7,500 rcf (Eppendorf Centrifuge 5804R). We then filter-sterilised supernatants by passing them through a 0.22 μm PES membrane filter and kept them at −20 °C. Meanwhile, we grew the receivers in 1 ml LB in 24-well plates for 24 h static at room temperature. Then, 1.5 μl of receiver cultures were added to CAA with 200 μM 2,2′-dipyridyl without or with 20 μl of donor supernatant (total culturing volume was 200 μl for both conditions) in 96-well plates in threefold replication and grown for 24 h at room temperature in a microplate reader. Growth trajectories were measured and analysed as described above. For each donor-receiver pair, we calculated the relative growth effect as the ratio of growth integrals of receivers with vs. without donor supernatant, and log-transformed this value to obtain normally distributed residuals. Values above or below zero indicate whether receivers were respectively stimulated or inhibited by the donor supernatant.

**Pyoverdine cross-feeding assay.** To test whether it is pyoverdine that triggers the above-observed growth effects, we performed cross-feeding assays using eight strain triplets (Table 1). We chose eight receivers, which produced less than 5% of pyoverdine compared to our reference strains. Each receiver originated from a different community (four soil and four pond communities). For each receiver, we picked two pyoverdine producers, which were previously shown to either stimulate or inhibit the growth of receivers in the supernatant assay. For one receiver, there was no inhibitor, and we thus chose a pyoverdine producer, supernatant of which had a neutral effect. We adapted the method of Meyer et al.[67] to crudely purify pyoverdine from the 16 producers (Supplementary Methods). For the cross-feeding assay, we suspended 3 mg of each purified pyoverdine in 500 μl Milli-Q water and passed the solution through a 0.22 μm PES filter. A 4 μl aliquot of the sterile pyoverdine solution was then added to 196 μl of iron-limited CAA (with 200 μM 2,2′-dipyridyl) in a 96-well plate, and inoculated with 1.5 μl of receiver LB overnight cultures in fivefold replication. As a control, we grew receivers without pyoverdine supplementation. Plates were kept for 24 h in a microplate reader. Growth trajectories were measured and analysed as described above.

**Competition assays.** We carried out pairwise competition assays between non-producers and their stimulatory or non-stimulatory pyoverdine producers under iron-limited conditions. To be able to distinguish the two competing strains, we used the mini-Tn7 system[68] to integrate a constitutively expressed mCherry marker into the chromosome of non-producers. Electroporation and conjugation protocols were adapted from Choi and Schweizer[68] (Supplementary Methods and Supplementary Table 4). To obtain growth profiles of the 24 strains (Fig. 5a) in iron-limited medium, we first grew mCherry-tagged non-producers and untagged producers in LB overnight. Then we adjusted cultures to $OD_{600} = 1$ with LB and added 2 μl of the adjusted cultures to 200 μl of CAA with 200 μM 2,2′-dipyridyl in 96-well plates, in triplicates. $OD_{600}$ was measured every 30 min at room temperature (25–28 °C) for 48 h in a Tecan microplate reader. Growth curves were analysed as described above (in the section 'Measurement of growth and pyoverdine production levels'). Prior to competition, we adjusted overnight LB cultures of all strains to $OD_{600} = 1$ in LB, and mixed strains in appropriate

volumetric ratios to obtain a roughly 1:1 starting frequencies (effective starting frequencies of non-producers varied between 0.27 and 0.50). A 2 μl aliquot of the strain mix was added to 200 μl iron-limited CAA (with 200 μM 2,2′-dipyridyl) in 96-well plates, where strains competed for 48 h (static incubation 25 C°) in six replicates. Following competition, the cultures were appropriately diluted in 0.85% NaCl, plated on LB-iron plates, and incubated at room temperature for 72 h. After 24–48 h, we counted total CFUs, and after 72 h, we used fluorescence imaging (Infinity3 camera system, Lumenera Corporation) to distinguish mCherry-tagged from untagged colonies using ImageJ (Supplementary Table 3). We calculated the relative fitness of the non-producer as $w_r = (a_t \times (1 - a_0))/(a_0 \times (1 - a_t))$, where $a_0$ and $a_t$ are the initial and final frequency of the non-producer, respectively[69]. Subsequently, we log-transformed $w_r$, such that values of $\ln(w_r) > 0$ or $\ln(w_r) < 0$ indicate whether non-producers have won or lost the competition, respectively.

**Genomic DNA isolation for whole-genome sequencing.** The 24 isolates were first grown on LB agar plates and then streaked out on Gould's S1 agar medium (supplemented with 40 μM FeCl₃) and grown for two days at 28 °C. Single colonies were inoculated into 4 ml LB in 14 ml polypropylene round-bottom tubes and grown overnight at 30 °C shaken (200 rpm). Genomic DNA was extracted from 1 ml of cultures using GenElute™ Bacterial Genomic DNA Kit (Sigma-Aldrich, Switzerland) according to the manufacturer's instructions, without RNase treatment. The main changes: we used a separately purchased proteinase K (20 mg/ml; Fermentas), and eluted DNA with EB elution buffer (Qiagen).

**Whole-genome sequencing and bioinformatic analysis.** Library preparation and sequencing of the 24 isolates was done at the Lausanne Genomic Technologies Facility with the Illumina Hiseq 2500 platform (paired end 2 × 250 bp) in Rapid Run mode. Libraries were constructed using the Truseq Nano DNA kit. Reads were filtered with Trimmomatic (Supplementary Table 3) and assembled with SPAdes 3.10.1 (Supplementary Table 3) using default parameters. Identification of putative coding sequences and their annotation were performed by the RAST automated annotation pipeline (Supplementary Table 3). The pyoverdine gene cluster was identified manually by a combination of keyword searches in the annotation and BLASTP searches with genes from the reference database against the genomes of isolates. In order to get gene-family profiles independent of annotation we also inferred an orthology using OrthoFinder (Supplementary Table 3). To compare the FpvA receptor encoded in the pyoverdine locus of the producers to receptors of the non-producers, the amino acid sequences of the FpvA receptor from the pyoverdine locus of the stimulators, and likewise the non-stimulators, were blasted against the genomes of the residual or complete non-producers. In the same way, the FpvA receptor sequence of residual and complete non-producers was compared to stimulating and non-stimulating producers. The best hits of the BLAST searches were then compared. The pyoverdine peptide structure was predicted from the amino acid sequence of non-ribosomal peptide synthetases by using PKS/NRPS analysis website (Supplementary Table 3).

**Statistical analysis.** We used linear and LMM models for statistical data analysis. All statistical tests are two-tailed, and $p$-values ≤ 0.05 were regarded as significant. Since strains isolated from the same community might not be independent from one another we built community identity as a random factor into our models. For our analysis on donor and receiver effects, we further added the strain ID of the donor as a random factor to our models. Whenever appropriate, we log-transformed (natural logarithm) data to meet the assumption of normally distributed residuals. For phylogenetic analysis we used the R package APE v3.2 (Supplementary Table 3). Blomberg's $K$-values were calculated with the phylosignal function from the R package picante v1.6-2 (Supplementary Table 3). All statistical analyses were carried out using R 3.1.2 program (www.r-project.org).

**Data availability.** The experimental and sequencing data that support the findings of this study have been deposited in the figshare repository (doi:10.6084/m9.figshare.5125093) and in the European Nucleotide Archive under the study accession number PRJEB21289 (http://www.ebi.ac.uk/ena), respectively.

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

## Acknowledgements

We thank Irene Fernández Delgado for help in the laboratory, Elisa Granato for help in the field, Adin Ross-Gillespie for statistical advice, Marta Pinto-Carbó for help with bioinformatic analyses and Leo Eberl, Gerardo Carcamo-Oyarce and Vladimir Sentchilo for providing strains. This work was funded by the Swiss National Science Foundation (PP00P3-139164 and PP00P3-165835) to R.K. and the Forschungskredit (FK-15-082) of the University of Zurich to E.B.

## Author contributions

E.B. and R.K. designed the study, E.B. isolated the strains and performed the experiments, M.B. and S.W. did the bioinformatic analyses, and all authors analysed the data and wrote the paper.

## Additional information

**Competing interests:** The authors declare no competing financial interests.

