## [transparent peer review file · Nature Communications]

Reviewer #1 (Remarks to the Author):

The manuscript by Butaite et al. describes the potential for social interactions among diverse *Pseudomonas* isolates from natural microbial communities.

This study focuses on the production of pyoverdinin, a secreted siderophore. The authors demonstrate that there is a continuum of pyoverdinin production among *Pseudomonas* isolates from soil and pond communities. Non-producers occur frequently and are widely distributed phylogenetically. The authors carry out a nice analysis of how different isolates can exploit (or not) pyoverdinin produced by other strains in the community, as has been shown with laboratory-evolved isolates.

The study is clear and well presented (for the most part, see some comments below), and represents a novel area of inquiry into the types of social interactions that can occur among natural isolates. Although the results are not surprising they are an important contribution to the field because they provide further support for the idea that complex social interactions occur among isolates from natural communities. However, the scope of the study is somewhat limited and therefore provides a narrow view – with some assumptions – about what is happening in these natural microbial communities. For example, see ‘major comments’ below.

Major comments:

Pages 8-9. One of the interesting parts of this manuscript is the discovery of what appears to be ‘inaccessible’ or ‘non-exploitable’ pyoverdins, which cannot be cheated on by certain strains of *Pseudomonas*. These ‘loner’ or ‘diverse siderotypes’ have been reported previously (see ref. 26 in this study and references therein). However, this study does not validate that the non-cheating phenotype they observe is in fact due to the diversity of siderophores produced by their isolates. There is also little discussion of how the diverse siderophore types are related phylogenetically. An analysis of the diversity of siderophores and the phylogeny of the strains that produce these, and how these strains relate to the non-exploitative, non-producing strains could be informative.

Another interesting discovery is that there is a continuum of pyoverdinin production among isolates. What accounts for this (at the molecular level)? It seems that there is enough known from the literature to do a molecular analysis of some of the strains to get an idea. It seems that if the partial producers are true cheaters then they will be closely related to full producer types and there will be only one or a few mutations that give rise to the partial producing phenotype.

Minor comments:

1. In the abstract, the sentence, “While some non-producers were cheats, by outcompeting producers through pyoverdine exploitation, some producers were resistant to cheating, by secreting pyoverdine types inaccessible to non-producers” is confusing. At the very least, this run-on sentence needs to be split into two sentences.
2. page 5, line184. It is not immediately evident what CAA medium is, and it’s importance.
3. Page 6, line 115. It is not immediately evident why rpoD was used for the basis of the phylogenetic trees.
4. page 8, line178. Here, and on the next page, the inaccessible pyoverdines are described as ‘non-exploitable’ whereas in the Fig. 5 legend they are described as inaccessible – this is confusing.
5. Page 9, line 202. It is not clear what ‘this producer’ means or what data this sentence is referring to.
6. Fig. 3 – what does the solid black line represent?
7. Fig. 5 – in several places ‘secrete’ was misspelled as ‘secret.’

Reviewer #2 (Remarks to the Author):

The authors investigate pyoverdine production and cross-feeding interactions among soil and pond isolates of *Pseudomonas*. Iron acquisition in *Pseudomonas* is a model for the study of microbial social behavior, and to expand research from the laboratory to the natural environment is an

important step. Specifically, the authors show that (taken from page 10 of the manuscript): (i) strains that produce no or low amounts of pyoverdine commonly occur in natural communities, although iron is a key growth-limiting factor for pseudomonads in most natural habitats; (ii) non-producers and low-producers often rely on pyoverdine produced by other community members to overcome iron limitation; (iii) certain non-producers can cheat: they outcompete producers by exploiting their pyoverdine; (iv) some producers secrete exclusive pyoverdine types that repress rather than promote the growth of non-producers; and (v) these interactions repeatedly occurred in replicated populations from two completely different habitats (pond and soil).

General comment

I feel a bit conflicted about this paper. On the one hand, I find it to be well-written, clearly structured, and for the most part, technically sound. On the other hand, I find the results and conclusions drawn to be of such a general nature that they do little more than confirm what is already known. In particular, it is already established that:

i. Pvd non-producers exist in natural communities (Meyer et al. Appl. Environ. Microbiol. 2002, Andersen et al. PNAS 2015, deVos et al. Arch Microbiol 2001)

ii. Environmental Pseudomonas strains are able to utilize heterologous Pvds and other siderophores, a phenomenon originally termed cross-feeding (shown in numerous studies, e.g. Hartney and Loper, Biometals 2011; Matthijs et al. Biometals 2009; Cornelis and Matthijs, 2002).

iii. Non-producers cheat on producers in natural communities (Pvd: Andersen et al. PNAS 2015; other siderophores: Cordero et al. PNAS 2012).

iv. There is enormous diversity in Pvd types that renders them exclusive to cells expressing a matching receptor (e.g. Cornelis and Matthijs, Env. Microbiol. 2002). Thus, it is inevitable to isolate a subset of Pseudomonas strains that cannot cross-feed on the Pvd type produced by another strain or species, resulting in growth suppression (e.g. Raaijmakers et al. Can. J. Microbiol. 1995).

Specific comments

1. I believe that a genotypic characterization of pvd-deficient isolates would greatly contribute to the impact of the study. Are these isolates Pvd-deficient because they lack biosynthesis genes, receptor genes, or both? It is hard to tell from phenotypic data alone. For example in *P. aeruginosa*, mutants either lacking Pvd biosynthesis genes or lacking the Pvd receptor FpvA show greatly reduced Pvd production. It would be of interest to know whether receptor genes of the homologous Pvd (often located in the same gene cluster) are preferentially retained, which would be indicative of social cheating/cross feeding as a predominant selective pressure. A more meaningful comparison with the Cordero et al. and Andersen et al. studies would be possible.

In addition, Fig. 3 shows that some pvd-deficient isolates can be cross-fed, while others cannot. A positive result indicates that the respective pvd-negative strains carry receptors that recognize heterologous Pvd. In contrast, a negative result does not tell us much, in part because pvd-negatives were only paired with a small subset of pvd-producing isolates. Do these isolates not have the ability to cross-feed due to a lack of Pvd receptors, or (presumably more likely) do they still have the potential to cross-feed, but the Pvd-positive strains just happen to produce an incompatible Pvd type?

2. Line 120: Is there no correlation between rpoD sequence diversity and geographical distance? It is not entirely clear from the text or from Fig. 2 whether the strain diversity within each soil or pond community (i.e. each environmental sample) is similar to that between different communities. This may well be the case, given the evidence from *P. aeruginosa* populations that local diversity is no different than global diversity (e.g. Pirnay et al. *Environ Microbiol* 2005; Kidd et al. *PLOS One* 2012). This type of population structure would suggest that it doesn't matter whether social interactions are investigated among local or global isolates, placing less emphasis on the specific sampling scheme employed.

3. Line 218: The conclusion that non-producers and low-producers often "rely" on pyoverdine produced by other community members to overcome iron limitation is probably an overstatement if it is meant to refer to the natural environment. First, we do not know whether producers and non-producers really interact in the natural communities sampled (soil isolates can be several millimeters apart), and second, we also do not know whether the pyoverdine-negative isolates produce other siderophores, albeit with an iron-binding affinity lower than that of the chelator added to the growth medium.

4. Line 445: Please provide more details on how bacterial growth is quantified and presented: I find it unusual to show growth as an integral. It is the authors' "parameter of interest", but why? Why not just use growth rate? In addition, how were "spline curves" fit to the growth trajectory? What was the growth model/equation used? The authors should provide additional details and cite appropriate literature.

5. The authors should include the Andersen et al. 2015 paper in their discussion, which shows Pvd cheating of *P. aeruginosa* in the CF lung.

Taken together, I find the study to be well written, methodically solid, and informative to researchers in the field, but given its largely confirmatory nature and the lack of genotypic data, I am not sure whether it has the novelty and depth that *Nature Communications* is looking for.

Reviewer #3 (Remarks to the Author):

Summary:

Butaite et al. show that the siderophore public good molecules mediate *Pseudomonas* bacterial social interactions and have clear consequences on the selection for cheating and cooperation in natural populations. Particularly that the diversity and extent of specificity of the siderophore molecule and receptor, therefore, the public good's shareability, leads to an antagonistic arms race and non-transitive dynamics.

General comments:

This is a nice paper written with clarity showing novel results for cheating-cooperator dynamics, outside the laboratory setting, depicting natural bacterial communities. The discussion section raises many interesting speculations as to what is happening in the natural world and the different explanations depending on how spatially structured the environment is, providing a lot of scope for future, more specific experimental testing.

One minor concern is the diversity at the genus level which the research is done. *Pseudomonas* species are very diverse that the variation of siderophores can be as vast as it would be if studying species across genera. Therefore, inevitably resulting in specificity across strains making the results less surprising.

While this research is novel, I think more supporting experiments would be useful. For example, assessing the variation of the pyoverdine molecule and receptors across the strains. How variable are the pyoverdine molecules and receptors? Do some strains in fact carry receptors for and produce multiple pyoverdine types or is it more common that they carry single types that are more broad range? This can be determined first by sequencing the pyoverdine locus to check for duplication and mutations. However, given that the data set consists of a large variation of species within *Pseudomonads*, the variation at the sequence level may also be vast and, therefore, hard to

interpret. A more concrete method would be to look at variation at the molecular level (peptide chains) and perhaps of a subset of the more phylogenetically closely related strains. Then extrapolate and speculate the extent of variation at a larger scale, assessing the role of this variation in inter-strain community dynamics.

Is the growth inhibition observed by some supernatants in fact due to bacteriocins in the media rather than just low affinity of pyoverdine? This can be easily tested for with a spot assay for a few of the strains in which inhibition was observed.

Specific comments:

Line 34: change "The perhaps" to "Perhaps the"

Line 38: change "focussed" to "focused"

Line 63-65: clarify sentence "Moreover some strains might produce exclusive pyoverdine types...producers themselves can use them." This won't confer resistance to cheating by isogenic pyoverdine mutants or simply lower producers at the clonal level.

Line 108: Clarify that iron has been added to the normal media or it otherwise it reads as though the media was simply not iron limited.

Line 146-147: Is there no correlation with growth and phylogenetic relatedness in pond communities because they are so much more diverse and therefore it is hard to pick up on this correlation? Perhaps clarify the reason why in the discussion

Line 150: clarify the sentence. How can "close relatives" be absent when relatedness is relative? What's the cut-off to be considered "close relative"? Do the authors mean relative to what is observed in the more structured soil communities?

Line 154: change "marginally" to "approaching"

Responses to the reviewers

Please find below the editorial and reviewer comments in bold and our specific responses in blue.

Yours truly,

Rolf Kümmerli (on behalf of all authors)

Editor's comments:

Your manuscript entitled "Siderophore cheating and competition for iron in natural *Pseudomonas* communities" has now been seen by three referees, whose comments are appended below. You will see from their comments copied below that while they find your work of considerable potential interest, they have raised quite substantial concerns that must be addressed. In light of these comments, we cannot accept the manuscript for publication, but would be interested in considering a revised version that addresses these serious concerns.

We hope you will find the referees' comments useful as you decide how to proceed. All three referees note that molecular characterization of the pyoverdine receptors across the *Pseudomonas* phylogeny would be necessary for this paper to comprise a novel and substantial advance over prior work. Should further experimental data or analysis allow you to address these criticisms, we would be happy to look at a substantially revised manuscript. However, please bear in mind that we will be reluctant to approach the referees again in the absence of major revisions. If the revision process takes significantly longer than three months, we will be happy to reconsider your paper at a later date, as long as nothing similar has been accepted for publication at Nature Communications or published elsewhere in the meantime.

Response 1: Many thanks for giving us the opportunity to revise our manuscript in the light of the reviewers' comments. We have paid great attention to carefully address all the comments (see detailed comments below). At this stage, we wish to briefly elaborate on the two main concerns raised by the referees:

- A) All three reviewers requested a molecular characterization of the pyoverdine synthesis cluster, the pyoverdine molecules, and the receptors. We have now performed such an extensive analysis by sequencing the entire genomes of the 24 natural *Pseudomonas* isolates, which we had used for the pyoverdine cross-feeding and competition assays. Thus, we can now provide detailed information on the molecular and genetic basis of the various social interactions reported in our paper, which significantly strengthens the conclusions of our work. Although we found the reviewers' request reasonable, we wish to stress that the molecular characterization was not a trivial endeavor. What might have escaped the attention of the reviewers is that pyoverdine is synthesized via non-ribosomal peptide synthesis. This means that there is no gene encoding the pyoverdine molecule. Instead, there is a large cluster of genes, which encode enzymes that build together the pyoverdine molecule in a complex assembly line. A further complication is that strains typically have multiple different pyoverdine receptors. Despite this complexity in both pyoverdine production and uptake, we could, thanks to sophisticated bioinformatics, resolve most of the reviewers' suggestions. Nonetheless, some of the requested analyses are simply not possible based on sequence data alone.

- B) Some reviewers think that our paper is rather of confirmatory nature and does not represent novel findings. We were surprised by such statements. It is true that there is a large body of work on siderophore production and uptake. However, we are not aware of any study that has quantified the actual fitness consequences of siderophore-mediated social interactions in natural communities. Moreover, many studies involved just a handful of strains, often non-systematically isolated from different habitats. Our study represents a great advance in this respect, as we studied interactions among co-occurring strains in replicated communities from two different habitats: soil and pond. Our work thus represents a general proof of the relevance of siderophore-mediated social interactions in natural communities.

Reviewers' comments:

Reviewer #1 (Remarks to the Author):

The manuscript by Butaite et al. describes the potential for social interactions among diverse *Pseudomonas* isolates from natural microbial communities.

This study focuses on the production of pyoverdinin, a secreted siderophore. The authors demonstrate that there is a continuum of pyoverdinin production among *Pseudomonas* isolates from soil and pond communities. Non-producers occur frequently and are widely distributed phylogenetically. The authors carry out a nice analysis of how different isolates can exploit (or not) pyoverdinin produced by other strains in the community, as has been shown with laboratory-evolved isolates.

The study is clear and well presented (for the most part, see some comments below), and represents a novel area of inquiry into the types of social interactions that can occur among natural isolates. Although the results are not surprising they are an important contribution to the field because they provide further support for the idea that complex social interactions occur among isolates from natural communities. However, the scope of the study is somewhat limited and therefore provides a narrow view – with some assumptions – about what is happening in these natural microbial communities. For example, see ‘major comments’ below.

Major comments:

Pages 8-9. One of the interesting parts of this manuscript is the discovery of what appears to be ‘inaccessible’ or ‘non-exploitable’ pyoverdins, which cannot be cheated on by certain strains of *Pseudomonas*. These ‘loner’ or ‘diverse siderotypes’ have been reported previously (see ref. 26 in this study and references therein). However, this study does not validate that the non-cheating phenotype they observe is in fact due to the diversity of siderophores produced by their isolates. There is also little discussion of how the diverse siderophore types are related phylogenetically. An analysis of the diversity of siderophores and the phylogeny of the strains that produce these, and how these strains relate to the non-exploitative, non-producing strains could be informative.

Response 2: The reviewer is correct that we did not demonstrate pyoverdinin diversity in our initial paper. We have now fixed this by predicting the peptide backbone of the pyoverdinin molecules produced by the 16 producers used in our pyoverdinin cross-feeding and competition assays (see new Table 1). Our analysis indeed reveals high pyoverdinin diversity and shows that, at the level of the community, the inaccessible pyoverdins are structurally different from the accessible pyoverdins.

We further agree that linking pyoverdine differences to phylogeny would be interesting. However, such an analysis is difficult (or even impossible) because pyoverdine differences do not necessarily correlate with sequence divergence at the genetic level, but is rather based on rearrangements in the non-ribosomal peptide synthesis assembly line (see our response 1). Thus, it is the order and sequence of amino acids in the pyoverdine backbone that changes, and this can vary dramatically between closely related strains without large genetic changes being involved (Smith et al. 2005 J. Bact. / Visca et al. 2007 Trends Microbiol). Currently, there is simply not enough known on these complex molecule assembly lines to perform the suggested analysis.

Another interesting discovery is that there is a continuum of pyoverdine production among isolates. What accounts for this (at the molecular level)? It seems that there is enough known from the literature to do a molecular analysis of some of the strains to get an idea. It seems that if the partial producers are true cheaters then they will be closely related to full producer types and there will be only one or a few mutations that give rise to the partial producing phenotype.

Response 3: This is a good point. Indeed, a lot is known about how pyoverdine is regulated in the human pathogen *Pseudomonas aeruginosa*. In this species, targeted mutagenesis of the *pvdS* gene (encoding the pyoverdine regulator) lead to a continuum in pyoverdine production (Wilson & Lamont 2006 J. Bact.). We are now citing and discussing these findings (lines 300-306).

Unfortunately, the exact regulation of pyoverdine in our natural isolates is not known, as several other elements, apart from *pvdS* likely play a role. Although all of our sequenced strains have a copy of *pvdS* (see new Table 1), and we found several sequence variations in this gene, we do not feel confident to claim that these variations are responsible for the continuum in the pyoverdine phenotype observed. This seems too speculative.

Minor comments:

1. In the abstract, the sentence, “While some non-producers were cheats, by outcompeting producers through pyoverdine exploitation, some producers were resistant to cheating, by secreting pyoverdine types inaccessible to non-producers” is confusing. At the very least, this run-on sentence needs to be split into two sentences.

Response 4: We have revised this part of the abstract.

2. page 5, line184. It is not immediately evident what CAA medium is, and it’s importance.

Response 5: We now explain what CAA medium is when mentioning it for the first time (on line 91)

3. Page 6, line 115. It is not immediately evident why *rpoD* was used for the basis of the phylogenetic trees.

Response 6: We now explain that *rpoD* is a housekeeping gene commonly used for phylogenetic classification of *Pseudomonas* strains (lines 87-89, 433). It is more variable than the 16S rRNA gene in this genus, and thus yields higher phylogenetic resolution.

4. page 8, line178. Here, and on the next page, the inaccessible pyoverdines are described as 'non-exploitable' whereas in the Fig. 5 legend they are described as inaccessible – this is confusing.

Response 7: We have standardized terminology and now consistently use the terms “compatible” or “incompatible” pyoverdines when referring to the pyoverdines that can or cannot be used by the non-producers, respectively.

5. Page 9, line 202. It is not clear what 'this producer' means or what data this sentence is referring to.

Response 8: We have rephrased this sentence and removed the above-mentioned ambiguity.

6. Fig. 3 – what does the solid black line represent?

Response 9: We now explain that the solid black line in Figure 3 depicts the significant positive correlation between the relative growth effect and the phylogenetic relatedness among soil isolates. Note also that we have split Figure 3 into two panels to make this complex data set more accessible to the readers.

7. Fig. 5 – in several places 'secrete' was misspelled as 'secret.'

Response 10: We have corrected this spelling mistake.

Reviewer #2 (Remarks to the Author):

The authors investigate pyoverdine production and cross-feeding interactions among soil and pond isolates of *Pseudomonas*. Iron acquisition in *Pseudomonas* is a model for the study of microbial social behavior, and to expand research from the laboratory to the natural environment is an important step. Specifically, the authors show that (taken from page 10 of the manuscript): (i) strains that produce no or low amounts of pyoverdine commonly occur in natural communities, although iron is a key growth-limiting factor for pseudomonads in most natural habitats; (ii) non-producers and low-producers often rely on pyoverdine produced by other community members to overcome iron limitation; (iii) certain non-producers can cheat: they outcompete producers by exploiting their pyoverdine; (iv) some producers secrete exclusive pyoverdine types that repress rather than promote the growth of non-producers; and (v) these interactions repeatedly occurred in replicated populations from two completely different habitats (pond and soil).

General comment

I feel a bit conflicted about this paper. On the one hand, I find it to be well-written, clearly structured, and for the most part, technically sound. On the other hand, I find the results and conclusions drawn to be of such a general nature that they do little more than confirm what is already known. In particular, it is already established that:

i. Pvd non-producers exist in natural communities (Meyer et al. Appl. Environ. Microbiol. 2002, Andersen et al. PNAS 2015, deVos et al. Arch Microbiol 2001)

ii. Environmental *Pseudomonas* strains are able to utilize heterologous Pvd's and other siderophores, a phenomenon originally termed cross-feeding (shown in numerous studies, e.g. Hartney and Loper, *Biometals* 2011; Matthijs et al. *Biometals* 2009; Cornelis and Matthijs, 2002).

iii. Non-producers cheat on producers in natural communities (Pvd: Andersen et al. *PNAS* 2015; other siderophores: Cordero et al. *PNAS* 2012).

iv. There is enormous diversity in Pvd types that renders them exclusive to cells expressing a matching receptor (e.g. Cornelis and Matthijs, *Env. Microbiol.* 2002). Thus, it is inevitable to isolate a subset of *Pseudomonas* strains that cannot cross-feed on the Pvd type produced by another strain or species, resulting in growth suppression (e.g. Raaijmakers et al. *Can. J. Microbiol.* 1995).

Response 11: We have already partly addressed this comment in our response 1B. We agree that our work builds on the papers mentioned above. But we disagree that our work is simply a confirmation of what is already known. Instead, our work makes several significant novel contributions. We wish to highlight three of them:

- A) Most of the earlier work on pyoverdine cross-use is based on qualitative assays. In Meyer et al. 1997 *Microbiology*, for instance, we find a table (Table 2) with (+), (\pm), and (0) describing the qualitative growth effects of different pyoverdines. This is different from our quantitative approach, which allows us to measure the actual fitness consequences of social interactions and relate them to the phylogenetic relatedness between strains.
- B) Most of the previous work on *Pseudomonas* involve a handful of strains often isolated from different locations. This is very much different from our replicated experimental design showing the importance of pyoverdine-mediated social interactions among co-occurring strains in both soil and pond communities. Replication is key in ecology to demonstrate the general relevance of a phenomenon.
- C) None of the above-mentioned studies has performed direct one-to-one competition assays, and thus nobody has demonstrated cheating among natural isolates so far. The above-mentioned papers by Cordero et al. 2012 and Andersen et al. 2015 did not demonstrate cheating. They simply found genetic patterns that are compatible with cheating. Here, we demonstrate that cheating can actually take place. We now explain this more clearly on lines 76-83.

Specific comments

1. I believe that a genotypic characterization of pvd-deficient isolates would greatly contribute to the impact of the study. Are these isolates Pvd-deficient because they lack biosynthesis genes, receptor genes, or both? It is hard to tell from phenotypic data alone. For example in *P. aeruginosa*, mutants either lacking Pvd biosynthesis genes or lacking the Pvd receptor FpVA show greatly reduced Pvd production. It would be of interest to know whether receptor genes of the homologous Pvd (often located in the same gene cluster) are preferentially retained, which would be indicative of social cheating/cross feeding as a predominant selective pressure. A more meaningful comparison with the Cordero et al. and Andersen et al. studies would be possible.

Response 12: Following the reviewer's suggestion, we have performed a genetic characterization of both the pyoverdine-deficient isolates and the pyoverdine producers used in our pyoverdine cross-

feeding and competition assays. The new data is provided in the Tables 1 and 2, and on lines 211-256. Here are some of the key new insights

- A) We found two types of non-producers: structural non-producers with a strongly truncated pyoverdine locus and silent non-producers with a complete, yet largely inactive locus (Table 1).
- B) All non-producers had multiple homologues of the *fpvA* receptor, one of which was part of the truncated pyoverdine locus (Table 2).
- C) Stimulating and non-stimulating producers indeed produce structurally different pyoverdines.
- D) Two complementary analyses revealed that receptor similarity was significantly higher between non-producers and producers secreting a growth-stimulating pyoverdine, than between non-producers and producers secreting a growth-inhibitory pyoverdine.

These insights are in perfect agreement with the fitness effects we show in Figure 4.

In addition, Fig. 3 shows that some pvd-deficient isolates can be cross-fed, while others cannot. A positive result indicates that the respective pvd-negative strains carry receptors that recognize heterologous Pvd. In contrast, a negative result does not tell us much, in part because pvd-negatives were only paired with a small subset of pvd-producing isolates. Do these isolates not have the ability to cross-feed due to a lack of Pvd receptors, or (presumably more likely) do they still have the potential to cross-feed, but the Pvd-positive strains just happen to produce an incompatible Pvd type?

Response 13: There is a misunderstanding here. All isolates used in our pyoverdine supplementation assay could cross-feed, but they could only take up the pyoverdine from certain producers, but not from others. Our new genetic analysis indeed confirms that all strains do possess one or several receptors in their genomes (Table 2). However, receptor similarity is significantly lower between non-producers and inhibiting producers than between non-producers and stimulating producers. The new molecular data combined with the data from our pyoverdine supplementation assay (Figure 4) strongly suggest that certain producers secrete an incompatible pyoverdine for which non-producers do not possess a matching receptor for uptake.

2. Line 120: Is there no correlation between *rpoD* sequence diversity and geographical distance? It is not entirely clear from the text or from Fig. 2 whether the strain diversity within each soil or pond community (i.e. each environmental sample) is similar to that between different communities. This may well be the case, given the evidence from *P. aeruginosa* populations that local diversity is no different than global diversity (e.g. Pirnay et al. Environ Microbiol 2005; Kidd et al. PLOS One 2012). This type of population structure would suggest that it doesn't matter whether social interactions are investigated among local or global isolates, placing less emphasis on the specific sampling scheme employed.

Response 14: We agree that investigating the relationship between genetic distance, geographical distance and behavioral patterns is of great interest. However, we believe that such analyses would go beyond the scope of the current paper, which is already dense in terms of the number of experiments and data presented. Our forthcoming paper will exactly focus on these types of questions.

3. Line 218: The conclusion that non-producers and low-producers often “rely” on pyoverdine produced by other community members to overcome iron limitation is probably an overstatement if it is meant to refer to the natural environment. First, we do not know whether producers and non-producers really interact in the natural communities sampled (soil isolates can be several millimeters apart), and second, we also do not know whether the pyoverdine-negative isolates produce other siderophores, albeit with an iron-binding affinity lower than that of the chelator added to the growth medium.

Response 15: The reviewer is correct, and we have toned down this argument.

4. Line 445: Please provide more details on how bacterial growth is quantified and presented: I find it unusual to show growth as an integral. It is the authors’ “parameter of interest”, but why? Why not just use growth rate? In addition, how were “spline curves” fit to the growth trajectory? What was the growth model/equation used? The authors should provide additional details and cite appropriate literature.

Response 16: We now explain why we used growth integrals (on lines 494-500). The essence is that spline fits are non-parametric fits, and thus there is no growth equation. Spline fits are used when bacterial cultures show non-standard (i.e. non-logistic) growth patterns, and growth cannot be described by standard models and metrics. The integral is useful in this context because it combines features of the entire growth kinetics into one single value. It is part of standard growth packages and often used in combination with spline fits (Kahme et al. 2010).

5. The authors should include the Andersen et al. 2015 paper in their discussion, which shows Pvd cheating of *P. aeruginosa* in the CF lung.

Response 17: We now refer to the Andersen et al. paper when discussing our sequencing results (lines 228-230). While we agree that Andersen et al. is an important paper, we must stress that this paper does not directly demonstrate cheating. It demonstrates a sequence of mutational events during the evolution of *P. aeruginosa* in the CF lung, which is compatible with the cheating hypothesis, but a direct demonstration of it is lacking.

Taken together, I find the study to be well written, methodically solid, and informative to researchers in the field, but given its largely confirmatory nature and the lack of genotypic data, I am not sure whether it has the novelty and depth that Nature Communications is looking for.

Response 18: We thank the reviewer for this overall positive assessment. Please also see our responses 1B and 11 regarding the seemingly confirmatory nature of our work.

Reviewer #3 (Remarks to the Author):

Summary:

Butaite et al. show that the siderophore public good molecules mediate *Pseudomonas* bacterial social interactions and have clear consequences on the selection for cheating and cooperation in natural populations. Particularly that the diversity and extent of specificity of

the siderophore molecule and receptor, therefore, the public good's shareability, leads to an antagonistic arms race and non-transitive dynamics.

General comments:

This is a nice paper written with clarity showing novel results for cheating-cooperator dynamics, outside the laboratory setting, depicting natural bacterial communities. The discussion section raises many interesting speculations as to what is happening in the natural world and the different explanations depending on how spatially structured the environment is, providing a lot of scope for future, more specific experimental testing.

Response 19: Many thanks for this positive evaluation.

One minor concern is the diversity at the genus level which the research is done. *Pseudomonas* species are very diverse that the variation of siderophores can be as vast as it would be if studying species across genera. Therefore, inevitably resulting in specificity across strains making the results less surprising.

Response 20: We agree that the genus *Pseudomonas* is very diverse. However, the overall genetic diversity does not necessarily correlate with pyoverdine divergence. This is because of the highly flexible and modular pyoverdine synthesis assembly line (see also our response 2 above). As a consequence, different strains from the same species can produce different pyoverdines (e.g. *P. aeruginosa*, Visca et al. 2007 Trends Microbiol.), whereas strains from different species can produce the exact same pyoverdine (see our new data in Table 1, e.g. isolates s3h14 and 3A7 produce the same pyoverdine although their *rpoD* identity is low, 90 %).

While this research is novel, I think more supporting experiments would be useful. For example, assessing the variation of the pyoverdine molecule and receptors across the strains. How variable are the pyoverdine molecules and receptors? Do some strains in fact carry receptors for and produce multiple pyoverdine types or is it more common that they carry single types that are more broad range? This can be determined first by sequencing the pyoverdine locus to check for duplication and mutations. However, given that the data set consists of a large variation of species within Pseudomonads, the variation at the sequence level may also be vast and, therefore, hard to interpret. A more concrete method would be to look at variation at the molecular level (peptide chains) and perhaps of a subset of the more phylogenetically closely related strains. Then extrapolate and speculate the extent of variation at a larger scale, assessing the role of this variation in inter-strain community dynamics.

Response 21: We have sequenced the whole genome of 24 isolates, and carried out all the suggested analyses (see our detailed responses 1A, 2, 12). The new data are presented in Tables 1 and 2, on lines 211-256. We further revised the discussion and implemented the novel insights gained from the molecular analyses. These analyses have significantly improved our understanding of the pyoverdine-mediated social interactions presented in our paper.

Is the growth inhibition observed by some supernatants in fact due to bacteriocins in the media rather than just low affinity of pyoverdine? This can be easily tested for with a spot assay for a few of the strains in which inhibition was observed.

Response 22: We agree that toxins, instead of pyoverdine, could explain some of the inhibitory effects observed in the supernatant assays. To rule out this possibility, we purified pyoverdines from eight producers that had either a growth-inhibiting or a neutral effect in the supernatant assays and tested whether the same effects are observed when pyoverdine alone was supplemented. This was indeed the case: seven of the eight siderophores were inhibitory (Figure 4), and none stimulated growth. This strongly suggests that it is mainly pyoverdine that causes the observed growth inhibitions.

Nonetheless, we agree that the involvement of bacteriocins cannot completely be ruled out and we discuss this possibility on lines 297-299. To further address the reviewer's comment, we have carried out an extra experiment. If we assume that the inhibition effects we report in Figure 3 were solely explained by bacteriocins and not by incompatible pyoverdines, then there should be a positive correlation between the strength of inhibition by the same supernatant in iron-poor versus iron-replete media (where pyoverdine is not needed). We tested this prediction for 64 strains, and found no significant correlation between the two tests (LMM: $t_{55} = 1.03$, $p = 0.31$, see figure below). This suggests that bacteriocins alone cannot explain the results in Fig. 3. If the editor and the reviewer recommend, we could add the extra analysis to the supplementary material of the paper. We prefer to leave it out because Fig. 4 already demonstrates that it is pyoverdine that is responsible for the observed inhibitory effects.

Specific comments:

Line 34: change “The perhaps” to “Perhaps the”

Response 23: Changed as proposed.

Line 38: change “focussed” to “focused”

Response 24: Changed as proposed.

Line 63-65: clarify sentence “Moreover some strains might produce exclusive pyoverdine types...producers themselves can use them.” This won't confer resistance to cheating by isogenic pyoverdine mutants or simply lower producers at the clonal level.

Response 25: We have now clarified that the scenario described here confers resistance to cheating by other non-isogenic strains (lines 64-67).

Line 108: Clarify that iron has been added to the normal media or it otherwise it reads as though the media was simply not iron limited.

Response 26: We have clarified this statement and specify that we added 40 μM iron chloride to CAA for this experiment (lines 106-109).

Line 146-147: Is there no correlation with growth and phylogenetic relatedness in pond communities because they are so much more diverse and therefore it is hard to pick up on this correlation? Perhaps clarify the reason why in the discussion

Response 27: Yes, indeed, phylogenetic diversity is larger in pond than in soil communities. In our initial version of the paper, we discussed an evolutionary scenario explaining why there is a correlation between growth stimulation and phylogenetic relatedness in soil but not pond communities on lines 149-153. Our line of argumentation matches the reviewer's point of view.

Line 150: clarify the sentence. How can "close relatives" be absent when relatedness is relative? What's the cut-off to be considered "close relative"? Do the authors mean relative to what is observed in the more structured soil communities?

Response 28: We agree that we cannot speak of "close relatives" as this would imply that we used a cut-off value. We have now clarified (lines 149-153) that we compare the values between strain pairs in soil versus pond communities.

Line 154: change "marginally" to "approaching"

Response 29: Changed as proposed (line 156).

Reviewer #1 (Remarks to the Author):

The revised manuscript addresses all of the concerns raised by this reviewer and is a much stronger manuscript in its current form. Three VERY minor suggestions were noted:

1. bottom of page 4, lines 82-83: "how the genetic and molecular basis of siderophore cheating and competition looks like" should be "what the genetic and molecular basis..."
2. Page 9, line 208: "worst" should be "worse"
3. Page 12, line 283: "do not" should be "not" (remove 'do')

Reviewer #2 (Remarks to the Author):

The authors have addressed all of my concerns. The sequencing data are a valuable addition to the paper. I have no further suggestions.

Reviewer #3 (Remarks to the Author):

I am satisfied with the author's response to reviewers and endorse publication of this manuscript.

Response to reviewer comments

Reviewer #1 (Remarks to the Author):

The revised manuscript addresses all of the concerns raised by this reviewer and is a much stronger manuscript in its current form. Three VERY minor suggestions were noted:

1. bottom of page 4, lines 82-83: “how the genetic and molecular basis of siderophore cheating and competition looks like” should be “what the genetic and molecular basis...”

We have modified this sentence, following the editor's suggestions, who also provided comments on this section.

2. Page 9, line 208: “worst” should be “worse”

Changed

3. Page 12, line 283: “do not” should be “not” (remove ‘do’)

Changed

Reviewer #2 (Remarks to the Author):

The authors have addressed all of my concerns. The sequencing data are a valuable addition to the paper. I have no further suggestions.

Reviewer #3 (Remarks to the Author):

I am satisfied with the author's response to reviewers and endorse publication of this manuscript.